# Comprehensive Examination of Unrolled Networks for Solving Linear Inverse Problems [note 1]

**DOI:** 10.3390/e27090929

**Published:** 2025-09-03

**Authors:** Yuxi Chen, Xi Chen, Arian Maleki, Shirin Jalali

**Affiliations:** 1Department of Statistics and Data Science, Carnegie Mellon University, Pittsburgh, PA 15213, USA; ericc3@andrew.cmu.edu; 2Electrical and Computer Engineering Department, Rutgers University, New Brunswick, NJ 08854, USA; xi.chen15@rutgers.edu; 3Department of Statistics, Columbia University, New York, NY 10027, USA; arian@stat.columbia.edu

**Keywords:** compressed sensing, deep unrolled networks, computational imaging

## Abstract

Unrolled networks have become prevalent in various computer vision and imaging tasks. Although they have demonstrated remarkable efficacy in solving specific computer vision and computational imaging tasks, their adaptation to other applications presents considerable challenges. This is primarily due to the multitude of design decisions that practitioners working on new applications must navigate, each potentially affecting the network’s overall performance. These decisions include selecting the optimization algorithm, defining the loss function, and determining the deep architecture, among others. Compounding the issue, evaluating each design choice requires time-consuming simulations to train, fine-tune the neural network, and optimize its performance. As a result, the process of exploring multiple options and identifying the optimal configuration becomes time-consuming and computationally demanding. The main objectives of this paper are (1) to unify some ideas and methodologies used in unrolled networks to reduce the number of design choices a user has to make, and (2) to report a comprehensive ablation study to discuss the impact of each of the choices involved in designing unrolled networks and present practical recommendations based on our findings. We anticipate that this study will help scientists and engineers to design unrolled networks for their applications and diagnose problems within their networks efficiently.

## 1. Unrolled Networks for Linear Inverse Problems

In many imaging applications, ranging from magnetic resonance imaging (MRI) and computational tomography (CT scan) to seismic imaging and nuclear magnetic resonance (NMR), the measurement process can be modeled in the following way:$$y=Ax^{*}+w.$$
In the above equation, $y\in R^{m}$ represents the collected measurements and $x^{*}\in R^{n}$ denotes the vectorized image that we aim to capture. The matrix $A\in R^{m\times n}$ represents the forward operator or measurement matrix of the imaging system, which is typically known exactly or with some small error. Finally, *w* represents the measurement noise, which is not known, but some information about its statistical properties (such as the approximate shape of the distribution) may be available.

Recovering $x^{*}$ from the measurement *y* has been extensively studied, especially in the last 15 years after the emergence of the field of compressed sensing [1,2,3,4]. In particular, between 2005 and 2015, many successful algorithms were proposed to solve this problem, including Denoising-based AMP [5,6], Plug-and-Play Priors [7], compression-based recovery [8,9], and Regularization by Denoising [10].

Inspired by the successful application of neural networks, many researchers have started exploring the application of neural networks to solve linear inverse problems [11,12,13,14,15,16,17,18,19,20,21,22,23,24,25,26,27,28,29,30,31,32,33,34,35,36,37,38]. The original networks proposed for this goal were deep networks that combined convolutional and fully connected layers [12,39]. The idea was that we feed *y* or $A^{T}y$ to a network and then expect the network to eventually return $x^{*}$.

While these methods performed reasonably well in some applications, in many cases, they underperformed the more classical non-neural network-based algorithms and simultaneously required computationally demanding training. Some of the challenges that these networks face are as follows:**Size of the measurement matrix**: The forward model under matrix *A* has m×n elements. Even if the image is small, we may have n=256×256 and m=128×128. This means that the measurement matrix may have more than 1 billion elements. Consequently, an effective deep learning-based recovery algorithm may need to memorize the elements or learn the structural properties of *A* to be able to reconstruct $x^{*}$ from *y*. This means that the neural network itself should ideally have many more parameters. Not only is the training of such networks computationally very demanding but also, within the computational limits of the work that has been conducted so far, end-to-end networks have not been very successful.**Changes in the measurement matrix**: Another issue that is faced by such large networks is that usually one needs to redesign a network and train a model specific to each measurement matrix. Each network often suffers from poor generalizability to even small changes in the matrix entries.**Forward model inconsistency**: It should also be noted that these end-to-end neural networks do not solve the inverse problem in the mathematical sense—they learn approximate mappings without guaranteeing consistency with the forward model y=Ax, as demonstrated in CT imaging applications [40].

To address the issues faced by such deep and complex networks in solving inverse problems, and inspired by iterative algorithms for solving convex and non-convex problems, a category of networks known as unrolled networks has emerged [41,42]. To understand the motivation behind these unrolled networks, we consider the hypothetical situation in which all images of interest belong to a set $C\subset R^{n}$. Under this assumption, one way to recover the image $x^{*}$ from the measurements *y* is to find(1)$$\begin{array}{c} \arg\min_{x\in C}{∥y-Ax∥}_{2}^{2}. \end{array}$$

One method to solve this optimization problem is via projected gradient descent (PGD), which uses the following iterative steps:(2)$$\begin{array}{ccc} {\tilde{x}}^{i} & = & x^{i}+\mu A^{T}\left(y-Ax^{i}\right),\\ x^{i+1} & = & P_{C}\left({\tilde{x}}^{i}\right). \end{array}$$
where $x^{i}$ is the estimate of $x^{*}$ in iteration *i*, μ is the step size (learning rate), and $P_{C}$ denotes the projection onto the set C. Figure 1 shows a diagram of the projected gradient descent algorithm.

One of the challenges in using PGD for linear inverse problems is that the set C is unknown, and hence $P_{C}$ is also not known. For example, C can represent all natural images of a certain size and $P_{C}\left(\cdot \right)$ a projection onto that space. Researchers have considered ideas such as using state-of-the-art image compression algorithms and image denoising algorithms for the projection step [6,7,8,9,10,43], and have more recently adopted neural networks as a promising approach [11,18,44]. In these formulations, all the projectors in Figure 1 are replaced with neural networks (usually, the same neural network architecture can be used at different steps). There are several analytical and computational benefits to such an approach:We do not require a heuristically pre-designed denoiser or compression algorithm to act as the projector $P_{C}\left(\cdot \right)$. Instead, we can deploy a training set to train the networks and optimize their performance. This enables the algorithm to potentially achieve better performance.Although projected gradient descent analytically employs the same projection operator at every iteration, once we replace them with neural networks, we do not need to impose the constraint of all the neural networks’ learned parameters being the same. In fact, giving more freedom can enable us to train the networks more efficiently and, at the same time, improve performance.Using neural networks enables greater flexibility and can integrate with a wide range of iterative optimization algorithms. For instance, although the above formulation of the unrolled network follows from projected gradient descent, one can also design unrolled networks using a wide range of options, including heavy-ball methods and message passing algorithms.

The above formulation of combining projected gradient descent with neural networks belongs to a family called *deep unrolled networks* or *deep unfolded networks*, which is a class of neural network architectures that integrate iterative model-based optimization algorithms with data-driven deep learning approaches [29,42,45]. The central idea is to “unroll” an iterative optimization algorithm a given number of times, where each iteration replaces the traditional mapping or projection operator with a neural network. The parameters of this network are typically learned end to end. By enabling parameters to be learnable within this framework, unrolled networks combine the interpretability and convergence properties of traditional algorithms with the adaptivity and performance of deep learning models. Due to their efficacy, these networks have been widely used in solving linear inverse problems.

The unrolled network can also be constructed by replacing the projected gradient descent algorithm with various other iterative optimization algorithms. Researchers have explored incorporating deep learning-based projectors with a range of iterative methods, including the Alternating Direction Method of Multipliers (ADMM-Net) [11], Iterative Soft-Thresholding Algorithm (ISTA-Net) [17], Nesterov’s Accelerated First Order Method [46,47], and approximate message passing (AMP-Net) [48]. Many of these alternatives offer faster convergence for convex optimization problems, raising the prospect of reducing the number of neural network projectors required, thereby lowering the computational complexity of both training and deploying these networks.

The remainder of this paper is organized as follows. Section 2 discusses the challenges in designing unrolled networks. Section 3 introduces our Deep Memory Unrolled Network (DeMUN), which generalizes existing unrolled algorithms. Section 4 presents our four main hypotheses with supporting experiments on loss functions, residual connections, and network complexity. Section 5 demonstrates robustness across different measurement matrices, noise levels, and image resolutions. Section 6 concludes with practical guidelines for designing effective unrolled networks.

## 2. Challenges in Using Unrolled Networks

As discussed above, the flexibility of unrolled networks has established them as a powerful tool for solving imaging inverse problems. However, applying these architectures to address specific inverse problems presents significant challenges to users. These difficulties stem primarily from two factors: (i) a multitude of design choices and (ii) robustness to noise, measurement matrix, and image resolution. We clarify these issues and present our approach to addressing them below.

### 2.1. Design Choices

The first challenge lies in the numerous design decisions that users must make when employing unrolled networks. We list some main choices below:**Optimization Algorithm**. In training any unrolled network, the user must decide on which iterative optimization algorithm to unroll. The choices include projected gradient descent, heavy-ball methods (such as Nesterov’s accelerated first-order method), approximate message passing (AMP), the Alternating Direction Method of Multiplies (ADMM), among others. Unrolled networks trained on different optimization algorithms may lead to drastically varying performances for the task at hand.**Loss Function**. For any unrolled optimization algorithm, given any observation *y*, one produces a sequence of *T* projections ${\left\{x^{i}\right\}}_{i=1}^{T}$ (see Figure 1 for an illustration). To train the model, the convention is to define the loss function with respect to the final projection $x^{T}$ using the ${𝓁}^{2}$ loss $∥x^{T}-x^{*}{∥}_{2}^{2}$ since this is usually the quantity returned by the network. However, given the non-convexity of the cost function that is used during training, there is no guarantee that the above loss function is optimal for the generalization error. For example, one could use a loss function that incorporates one or more estimates from intermediate stages, such as $∥x^{T}-x^{*}{∥}_{2}^{2}+\;{∥x^{T/2}-x^{*}∥}_{2}^{2}$, to potentially achieve better training that provides an improved estimate of $x^{*}$. As will be shown in our simulations, the choice of the loss function has a major impact on the performance of the learned networks. Various papers have considered a wide range of loss functions for training different networks. We categorize them broadly below.**–** **Last-Layer Loss**. Consider the notations used in Figure 1. The last-layer loss evaluates the performance of the network using the following loss function:(3)$$\begin{array}{c} {𝓁}_{ll}\left(x^{T},x^{*}\right)={∥x^{T}-x^{*}∥}_{2}^{2}. \end{array}$$The last-layer loss is the most popular loss function that has been used in applications. The main argument for using this loss is that, since we only care about the final estimate and that is used as our final reconstruction, we should consider the error of the last estimate.**–** **Weighted Intermediate Loss**. While the loss function above seems reasonable, some works in related fields have proposed using an intermediate loss function instead [49,50]. We define the general version of the intermediate loss function as follows:(4)$$\begin{array}{c} {𝓁}_{i,\omega }\left(x^{1},x^{2},\ldots ,x^{T},x^{*}\right)=\sum_{i=1}^{T}\omega^{T-i}\left|\right|x^{i}-x^{*}{\left|\right|}_{2}^{2}, \end{array}$$
where ω∈(0,1]. One argument that motivates the use of such loss function is that, if the predicted image after each projection is closer to the ground truth $x^{*}$, then it will help the subsequent steps to reach better solutions. The weighted intermediate loss tries to find the right balance among the accuracy of the estimates at different iterations [49]. In addition, we make the following observations:∗When ω=1, the losses from different layers of the unrolled network are weighted equally. This means that our emphasis on the performance of the last layer is “weakened.” However, this is not necessarily undesirable. As we will show in our simulations, improving the estimates of the intermediate steps will also help to improve the recovery quality of $x^{T}$.∗As we decrease the value of ω→0, we see that the loss function ${𝓁}_{i,\omega }$ approaches the last-layer loss ${𝓁}_{ll}$. The choice of the ω therefore enables us to interpolate between the two cases.**–** **Skip *L*-Layer Intermediate Loss**. Another loss function that we investigate is what we call the skip *L*-layer intermediate loss. This loss is similar to the loss used in Inception networks for image classification [51]. Let *L* be a factor of *T*. Then, the skip *L*-layer loss is given by(5)$$\begin{array}{c} {𝓁}_{s,L}\left(x^{1},x^{2},\ldots ,x^{T},x^{*}\right)=\sum_{i=0}^{T/L-1}{∥x^{T-iL}-x^{*}∥}_{2}^{2}. \end{array}$$For instance, if T=15 and L=3, the skip 3-layer intermediate loss will evaluate the sum of the mean-squared errors between $x^{*}$ and projections $x^{3},x^{6},x^{9},x^{12},$ and $x^{15}$. By ranging *L* from 1 to *T*, one can again interpolate between the two loss functions ${𝓁}_{i,1}$ and ${𝓁}_{ll}$.**Number of Unrolled Steps**. Practitioners also have to decide on the number of steps *T* to unroll for any optimization algorithm. Increasing *T* often comes with additional computational burdens and may also lead to overfitting. A proper choice of *T* can ensure that network training is not prohibitively expensive and ensure desirable levels of performance.**Complexity of the Neural Network**. Similar to the above, the choice of $P_{C}$ also has a significant impact on the performance of the network. The options entail the number of layers or depth of the network, the activation function to use, whether or not to include residual connections, etc. If the projector is designed to have only little capacity, the unrolled network may have poor recovery. However, if the projector has excessive capacity, the network may become computationally expensive to train and prone to overfitting.

It is important to note that, after making all the design choices, users are required to conduct time-consuming, computationally demanding, and costly simulations to train the network. Consequently, users may only have the opportunity to explore a limited number of options before settling on their preferred architecture.

### 2.2. Robustness and Scaling

When designing unrolled networks for inverse problems, it is common to aim for robustness across a range of settings beyond the specific conditions for which the algorithm was originally designed. While an algorithm may be tailored for a particular signal type, image resolution, number of observations, or noise level, it is desirable for the network to maintain effectiveness across different settings as well.

As a simple motivating example, consider when a new imaging device has been acquired that operates using a different observation matrix. If the unrolled network previously designed has bad adaptivity and performance with respect to the current matrix, one would be required to review the entire process to determine a new batch of choices for the current setting. Therefore, ideally, one would like to have a single network structure that works well across a wide range of applications.

### 2.3. Our Approach for Designing Unrolled Networks

As discussed above, one faces an abundance of design choices before training and deploying an unrolled network. However, testing the performance of all the possible enumerations of these choices across a wide range of applications and datasets is computationally demanding and combinatorially prohibitive. This hinders practitioners from applying the optimal unrolled network in their problem-specific applications. To offer a more systematic way for designing such networks, we adopt the following high-level approach:We present the **Deep Memory Unrolled Network (DeMUN)**, where each step of the network leverages the gradient from all previous iterations. These networks encompass various existing models as special cases. The DeMUN lets the data decide on the optimal choice of algorithm to be unrolled and improves recovery performance.We present several hypotheses regarding important design choices that underlie the design of unrolled networks, and we test them using extensive simulations. These hypotheses allow users to avoid exploring the multitude of design choices that they have to face in practice.

These two steps allow users to bypass many design choices, such as selecting an optimization algorithm or loss function, thus simplifying the process of designing unrolled networks. We test the robustness of our hypotheses with respect to the changes in the measurement matrices and noise in the system. These robustness results suggest that the simplified design approach presented in this paper can be applied to a much wider range of systems than those specifically studied here.

## 3. Deep Memory Unrolled Network (DeMUN)

As discussed previously, one of the initial decisions users face when designing an unrolled network is selecting the optimization algorithm to unroll. Various optimization algorithms, including gradient descent, heavy-ball methods, and approximate message passing, have been incorporated into unrolled networks. We introduce the Deep Memory Unrolled Network (DeMUN), which encompasses many of these algorithms as special cases. At the *i*-th iteration in the DeMUN, the update of ${\tilde{x}}^{i}$ is given by(6)$${\tilde{x}}^{i}=\alpha^{i}x^{i}+\sum_{j=0}^{i}\beta_{j}^{i}A^{T}\left(y-Ax^{j}\right),$$
for i∈{0,…,T−1}, where $x^{0}=0$. In other words, while calculating ${\tilde{x}}^{i}$, it uses not only the gradient calculated at the current step but also leverages all the gradients calculated from previous steps.

By using different choices for $\beta_{0}^{i},\beta_{1}^{i},\ldots ,\beta_{i}^{i}$ at each iteration, one can recover a large class of algorithms, including gradient descent, heavy-ball methods, and approximate message passing. As shown in Equation (Equation 6) and illustrated in Figure 2, we can rearrange the vector $x^{i}$ and the gradients ${\left\{A^{T}\left(y-Ax^{j}\right)\right\}}_{j\in \{0,\ldots ,i\}}$ as images and view the expression as one-by-one convolutions over the images. Our simulation results reported later show that DeMUNs with trainable $\beta_{0}^{i},\beta_{1}^{i},\ldots ,\beta_{i}^{i}$ offer greater flexibility and better performance compared to fixed instances such as gradient descent or Nesterov’s method.

## 4. Our Four Main Hypotheses

### 4.1. Simulation Setup

Our goal in this section is to (1) show the effectiveness of the DeMUN by comparing its performance against different unrolled algorithms and (2) explore the impact of specific design choices. We conduct extensive ablation studies where we fix all but one design choice at each step and explore the performance of unrolled algorithms under different options for this choice. Based on these studies, we have developed several hypotheses aimed at simplifying the design of unrolled networks. We will outline these hypotheses and present simulation results that support them.

For all simulations below, we report results of four different sampling rates m/n for the measurement matrix *A*: 10%, 20%, 30%, and 40%. In Section 4, each entry in the measurement matrix is i.i.d. Gaussian, where $A_{ij}∼N\left(0,1/m\right)$ for $A\in R^{m\times n}$. While we will discuss the impact of the resolution on the performance of the algorithms, in the initial simulations, all training images have resolution 50×50, and vectorizing the images leads to n=2500. We primarily consider when the number of unrolled steps T=30, with additional comparisons to performance at T=5,15, where illustrative. For all results below, we report the Peak Signal-to-Noise Ratio (PSNR) and the Structural Similarity Index Measure (SSIM) for the networks trained under the aforementioned sampling rates and number of projection steps on a test set of 2500 images. More details on data collection and processing, training of unrolled networks, and evaluation are deferred to Appendix A.

In our simulations, we adopt the general DnCNN architecture as outlined by Zhang et al. as our neural network projector $P_{C}$ [52]. A DnCNN architecture with *L* intermediate layers consists of an input layer with 64 filters of size 3×3×1 followed by an ReLU activation function to map the input image to 64 channels (It is of size 3×3×1 since we assume the images are in grayscale.), *L* layers consisting of 64 filters of size 3×3×64 followed by BatchNormalization and ReLU, and a final reconstruction layer with a single filter of size 3×3×64 to map to the output dimension of 50×50×1.

### 4.2. Overview of Our Simplifying Hypotheses

As previously described, we begin with four hypotheses, each of which contributes to improving the performance of unrolled networks, enhancing training practices, and simplifying the design process by reducing the number of decisions practitioners need to make. These hypotheses are based on extensive simulations and are reported below.

**Hypothesis** **1.**
*Unrolled networks trained with the loss function ${𝓁}_{i,1}$ uniformly outperform their counterparts trained with ${𝓁}_{ll}$. Among the unrolling algorithms we tested, i.e., PGD, AMP, and Nesterov, DeMUNs offer superior recovery performance.*


Although we are primarily concerned with the quality of the final reconstruction $∥x^{T}-x^{*}{∥}_{2}^{2}$, we find that using the loss function ${𝓁}_{i,1}=\sum_{i=1}^{T}\left|\right|x^{i}-x^{*}{\left|\right|}_{2}^{2}$ during training yields better recovery performance than focusing solely on the last layer. This improvement may be attributed to the smoother optimization landscape provided by using the intermediate loss, which guides the network more effectively towards better minima. We present our empirical evidence for suggesting this hypothesis in Section 4.3. With the advantage of using an unweighted intermediate loss function established, we next explore the impact of incorporating residual connections into unrolled networks.

**Hypothesis** **2.**
*DeMUNs trained using residual connections and loss function ${𝓁}_{i,1}$ uniformly improve recovery performance compared to those trained without residual connections.*


Residual connections are known to alleviate issues such as vanishing gradients and facilitate the training of deeper networks by allowing gradients to propagate more effectively through the intermediate layers [53,54]. Specifically, we modify each projection step in our unrolled network to be $x^{i+1}={\tilde{x}}^{i}+P_{C}\left({\tilde{x}}^{i}\right)$. In verifying Hypothesis 2, we continue to use ω=1 (see the definition of intermediate loss in (Equation 4)). This ensures that any observed improvements can be directly attributed to the addition of residual connections rather than changes in the loss function. We present our empirical evidence for suggesting this hypothesis in Section 4.4. Having confirmed that both the use of an unweighted intermediate loss and the inclusion of residual connections improve recovery performance, we further investigate the sensitivity of our network to the specific shape of the loss function.

**Hypothesis** **3.**
*For training DeMUNs, there is no significant difference among the following loss functions: (1) ${𝓁}_{i,1}$, (2) ${𝓁}_{i,0.95}$, and (3) ${𝓁}_{i,0.85}$. Furthermore, ${𝓁}_{i,0.5}$, ${𝓁}_{i,0.25}$, ${𝓁}_{i,0.1}$, ${𝓁}_{i,0.01}$, and ${𝓁}_{s,5}$ perform worse than ${𝓁}_{i,1}$.*


**Hypothesis** **4.**
*When we vary the number of layers, L, in the DnCNN from 5 to 15, the performance of DeMUNs remains largely unchanged, indicating that the number of layers has a negligible impact on its performance. However, increasing L from 3 to 5 provides a noticeable improvement in performance.*


Confirming these hypotheses provides a set of practical recommendations for designing unrolled networks that are both effective and robust across various settings.

### 4.3. Impact of Intermediate Loss

In this section, we aim to validate Hypothesis 1, which posits that deep unrolled networks trained with the unweighted intermediate loss function ${𝓁}_{i,1}$ uniformly outperform their counterparts trained with the last-layer loss ${𝓁}_{ll}$. We consider the following algorithms:Deep Memory Unrolled Network (DeMUN): Our proposed network that incorporates the memory of all the gradients into the unrolling process.Projected Gradient Descent (PGD): The standard unrolled algorithm outlined in (Equation 2).Nesterov’s Accelerated First-Order Method (Nesterov): An optimization method that uses momentum to accelerate convergence [46].Approximate Message Passing (AMP): An iterative algorithm tailored for linear inverse problems with Gaussian sensing matrices [6,15].

For all unrolled algorithms, we consider when all the projection steps are cast as direct projections of the form $x^{i+1}=P_{C}\left({\tilde{x}}^{i}\right)$ and compare the performance between last-layer loss and unweighted intermediate loss. Figure 3 presents an example of a DnCNN architecture with L=3 intermediate layers.

Regarding Table 1 and Figure 4 and Figure 5, we make the following remarks.


**Improved Performance with Intermediate Loss:**
By analyzing the tables and graphs, we conclude that, across all four unrolled algorithms, training with the intermediate loss function ${𝓁}_{i,1}$ consistently yields higher PSNR values compared to training with the last-layer loss, ${𝓁}_{ll}$.**Superiority of Deep Memory Unrolled Network:** Among all algorithms that we have unrolled, i.e., PGD, Nesterov, and AMP, the DeMUN achieves the highest PSNR values when trained with the intermediate loss, confirming our hypothesis. (However, we see that this is not always the case when using last-layer loss. A possible explanation is that our memory networks contain many parameters (especially with many projection steps) and may be stuck at a local minimum during training using the last-layer loss. In contrast, when adopting the intermediate loss function, the network needs to optimize its projection performance across all projection steps to minimize the loss. As a result, it may find better solutions, especially for the parameters that are involved in the earlier layers.) This is to be expected as DeMUNs encompass the other unrolled networks as special cases. During training, the data effectively determines which algorithm should be unrolled.

According to these observations, the intermediate loss may provide several benefits:**Avoiding Poor Local Minima:** Focusing solely on the output of the final layer may lead the network to suboptimal solutions (due to non-convexity). In comparison, the intermediate loss encourages the network to make meaningful progress at each step, which potentially reduces the risk of becoming stuck in poor local minima.**More Information during Backpropagation:** By including losses from all intermediate steps, the network receives more gradient information during autodifferentiation, which may be helpful in learning better representations and weights.

These empirical results strongly support our first hypothesis that incorporating information from all intermediate steps creates a more effective learning mechanism for the network.

### 4.4. Impact of Residual Connections

Having verified that training with the intermediate loss function ${𝓁}_{i,1}$ improves the recovery performance of unrolled networks, we now examine the effect of incorporating residual connections of the form $x^{i+1}={\tilde{x}}^{i}+P_{C}\left({\tilde{x}}^{i}\right)$ into unrolled networks, as stated in Hypothesis 2, when fixing the choice of the unweighted intermediate loss function ${𝓁}_{i,1}$. For comparison, in addition to the Deep Memory Unrolled Network, we include the results for unrolled networks based on PGD under the same conditions.

From Table 2 and Figure 6, we observe the following:**Consistent Performance Improvement:** Including residual connections consistently improves the PSNR across all sampling rates and number of projection steps for both the deep memory- and PGD-based unrolled networks.**Superior Performance of Deep Memory Network:** While both networks benefit from residual connections, the Deep Memory Unrolled Networks maintain superior performance over projected gradient descent in all scenarios.

These empirical results strongly support Hypothesis 2 that incorporating residual connections into the Deep Memory Unrolled Network further improves its performance on top of training with the unweighted intermediate loss function. The consistent improvement across different sampling rates and projection steps potentially highlights the value of residual connections in unrolled network architectures.

### 4.5. Sensitivity to Other Loss Functions

Having identified that using an unweighted intermediate loss function and incorporating residual connections in Deep Memory Unrolled Networks offer superior performance, we now explore the sensitivity of our network to variations in the loss function as raised in Hypothesis 3. Specifically, we want to see whether different weighting schemes in the intermediate loss function or using a skip-*L* layer loss significantly impact the recovery performance. We consider the following variations of the loss function: ${𝓁}_{i,\omega }$, where ω∈{0.95,0.85,0.75,0.5,0.25,0.1,0.01} and ${𝓁}_{s,5}$. We also include results for Deep Memory Unrolled Networks trained using ${𝓁}_{ll}$ with residual connections for comparison.

From the simulation results presented in Table 3, we are able to observe the following:**Minimal Impact for ω≥0.75:** When ω∈{1,0.95,0.85,0.75}, the recovery performance remains relatively consistent, with negligible differences in the PSNR values.**Degradation with Small ω:** For ω∈{0.5,0.25,0.1,0.01}, there is a noticeable decrease in reconstruction quality. This decline may be attributed to the exponential down-weighting of the initial layers, which causes the network to focus excessively on the later iterations, potentially leading to suboptimal convergence.

We see that as long as the intermediate outputs receive sufficient emphasis during training, the network can output high-quality reconstructions. The decline in performance with smaller values of ω underscores the importance of adequately supervising the reconstruction of intermediate layers to guide the network toward the desirable recovery.

### 4.6. Impact of the Complexity of the Projection Step

In this section, we examine Hypothesis 4 by changing the number of intermediate layers *L* of the DnCNN architecture. We assume that there is no additive measurement noise and consider L=3,5,10, and 15 layers. Our results are shown below.

We summarize our conclusions from Table 4, Table 5 and Table 6 below:Increasing the number of layers from 5 to 15 results in negligible changes in the performance of DeMUNs regardless of the number of projection steps. By comparing Table 4, Table 5 and Table 6, we observe that the number of projections has a significantly greater impact on performance than the number of layers within each projection.By comparing L=3 and L=5, we conclude that reducing the depth too drastically (L≤3) may impair the network’s ability to learn complex features as convolutional neural networks rely on multiple layers to capture hierarchical representations [55].

We acknowledge that these conclusions may not necessarily extend to other projector architectures that do not rely on deep convolutional layers. Nevertheless, we believe this observation generalizes to other types of architectures when their capacity diminishes beyond a certain threshold, although we defer further investigation to future work. We address extensions to other types of measurement matrices in Section 5.3.

## 5. Robustness of DeMUNs

In Section 4, we established, through extensive simulations, the superior performance of DeMUNs trained with unweighted intermediate loss ${𝓁}_{i,1}$ and residual connections. The aim of this section is to assess the robustness of this configuration under various conditions. Specifically, we examine our network’s performance under changes in the measurement matrix, the presence of additive noise, variations in input image resolution, and changes in projector capacity. These aspects represent the primary variables that practitioners must consider when deploying unrolled networks in real-world scenarios. Our extensive experiments demonstrate the adequacy and generalizability of our design choices. In the simulations presented in the following sections, we fix the image resolution to 50×50 when the resolution has not been specified.

### 5.1. Robustness to the Sampling Matrix

We first investigate our network’s performance under different sampling matrix structures. In addition to the Gaussian random matrix used previously, we consider a Discrete Cosine Transform (DCT) matrix of the form $A=SF\in R^{m\times n}$, where $S\in R^{m\times n}$ is an undersampling matrix and *F* represents the 2D-DCT. We set the number of hidden layers for each projector (DnCNN) L=5. Additional implementation details can be found in Appendix A. There are a few points that we would like to clarify here:Table 7 demonstrates that our network maintains good performance when considering DCT-type measurement matrices as well. The network effectively adapts to the DCT matrices, achieving comparable or better PSNR values than the Gaussian forward model. This suggests that our design choices made based on our simulations on Gaussian forward models offer good performance for other types of matrices as well.The performance improvement DeMUNs gain from additional projection steps on DCT forward models is typically less than the improvement achieved with additional projections on Gaussian matrices. Since there are no signs of overfitting concerning recovery performance, we believe that the user does not need to worry about the number of projection steps when designing the network.

### 5.2. Robustness to Additive Noise

Next, we introduce additive noise and obtain measurements of the form y=Ax+w, where $\omega ∼N(0,\sigma^{2}I)$. We want to see if our design choices still offer good performance in the presence of additive noise. The primary objective of this section is to demonstrate that the PSNR of DeMUN reconstructions gradually decreases as the noise level increases and overfitting does not occur as the number of projections increases.

We summarize some of our conclusions from Table 8 and Table 9 below:Despite additive measurement noise predictably lowering the recovery PSNR, its impact on performance is relatively controlled. In particular, as the noise level increases, the PSNR degrades at a rate significantly slower than the decrease in the input SNR. This suggests that the network effectively suppresses the measurement noise.As the noise level grows, the marginal benefit of additional projection steps diminishes. In other words, fewer projection steps often suffice to achieve comparable reconstruction quality. As mentioned before and is clear from Table 8, still increasing the number of projections does not hurt the reconstruction performance of the network. Hence, in scenarios where the noise level is not known, practitioners may choose a number that works well for the noiseless setting and use it for the noisy settings as well.

### 5.3. Robustness of Hypothesis 4 to Sampling Matrix and Additive Noise

The main goal of this section is to evaluate the robustness of Hypothesis 4 in response to changes in the measurement matrix and measurement noise. We first assume that there is no additive noise and consider L=3,5,10, and 15 layers. We then evaluate the performance of DeMUNs on DCT-type matrices described in Section 5.1.

As evident from Table 10, increasing *L* from 5 to 15 does not provide a noticeable improvement for DCT-type matrices. One could also argue that, in most cases for DCT-type matrices, the performance gain from increasing *L* from 3 to 5 is marginal.

Next, we study the accuracy of Hypothesis 4 when additive noise is present in the measurements. Here, we consider three noise levels σ∈{0.01,0.025,0.05} and test depths of L=3,5, and 10. The results are presented in Table 11 and Table 12.

These results strongly suggest that, even in the presence of additive noise, increasing *L* does not offer substantial gain in the performance of DeMUNs. Given that the improvement in recovery performance is marginal when increasing the projector capacity, this suggests that simple architectures like DnCNN with very few convolutional layers may be sufficient for practical applications where measurement noise is present, offering potential computational savings without significant performance degradation.

### 5.4. Robustness to Image Resolution

Finally, we assess the DeMUN’s performance across different image resolutions. We test resolutions of 32×32,50×50, 64×64, and 80×80, fixing the measurement matrices and removing measurement noise. There are two main questions we aim to address here: (1) Do we need more or fewer projections as we increase the number of projections? (2) How should we set the number of layers *L* in the projection as we increase/decrease resolution? As before, we first set the number of intermediate layers of each projector to L=5.

We observe from Table 13 that, as the image resolution increases, the network’s recovery performance generally improves. This is possibly due to the presence of more information in higher-resolution images, which helps the network in learning more detailed structural properties.

## 6. Conclusions

In this paper, we conducted a comprehensive empirical study on the design choices for unrolled networks in solving linear inverse problems. As our first step, we introduced the Deep Memory Unrolled Network (DeMUN), which leverages the history of all gradients and generalizes a wide range of existing unrolled networks. This approach was designed to (1) allow the data to decide on the optimal choice of algorithm to be unrolled and (2) improve recovery performance. A byproduct of our choice is that users do not need to decide which algorithm they need to unroll. Figure 7 presents examples of recovered images under DCT matrix with 30 projections across different sampling rates.

Through extensive simulations, we demonstrated that training the DeMUN with an unweighted intermediate loss function and incorporating residual connections represents the best existing practice (among the ones studied in this paper) for optimizing these networks. This approach delivers superior performance compared to existing unrolled algorithms, highlighting its effectiveness and versatility.

We also presented experiments that exhibit the robustness of our design choices to a wide range of conditions, including different measurement matrices, additive noise levels, and image resolutions. Hence, our results offer practical guidelines and rules of thumb for selecting the loss function for training, structuring the unrolled network, determining the required number of projections, and deciding on the appropriate number of layers. These insights simplify the design and optimization of such networks for a wide range of applications, and we expect them to serve as a useful reference for researchers and practitioners in designing effective unrolled networks for linear inverse problems across various settings.

## Figures and Tables

**Figure 1 entropy-27-00929-f001:**
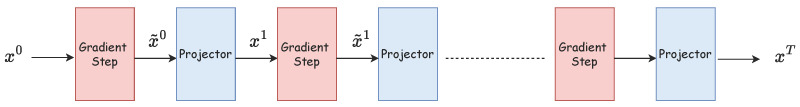
Diagram of projected gradient descent. Starting with $x^{0}=0$, the $i^{\mathrm{th}}$ gradient step performs the operation ${\tilde{x}}^{i}=x^{i}+\mu A^{T}\left(y-Ax^{i}\right)$, and the $i^{\mathrm{th}}$ projector unit performs $x^{i+1}=P_{C}\left({\tilde{x}}^{i}\right)$.

**Figure 2 entropy-27-00929-f002:**
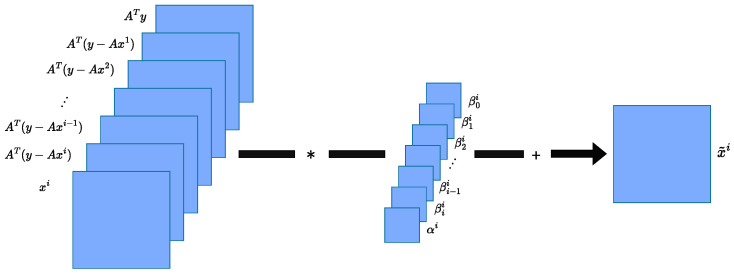
An example of the memory terms combined into a single image.

**Figure 3 entropy-27-00929-f003:**
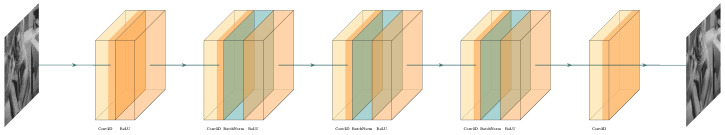
An example of the DnCNN architecture with L=3 intermediate layers.

**Figure 4 entropy-27-00929-f004:**
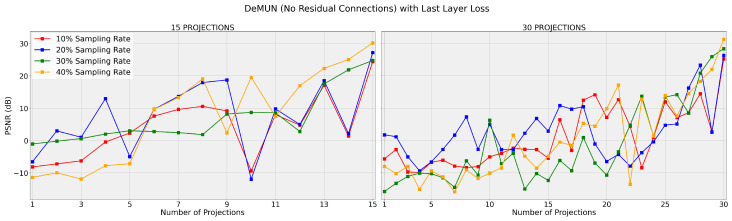
DeMUN (no residual connections) with loss ${𝓁}_{ll}$. The networks are trained for T=15 (left) and T=30 (right), and the graph displays the PSNR after each intermediate projection. Both graphs share the same y-axis scale.

**Figure 5 entropy-27-00929-f005:**
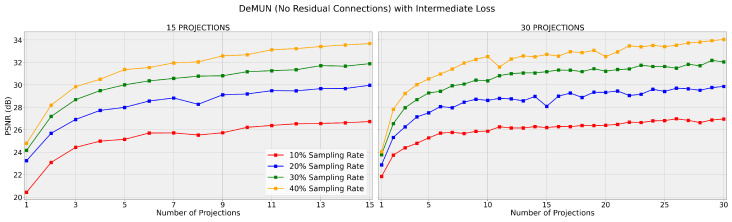
DeMUN (no residual connections) with loss ${𝓁}_{i,1}$. The networks are trained for T=15 (left) and T=30 (right), and the graph displays the PSNR after each intermediate projection. Both graphs share the same y-axis scale.

**Figure 6 entropy-27-00929-f006:**
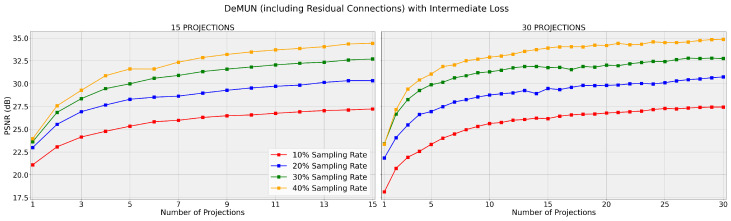
DeMUN (including residual connections) with loss ${𝓁}_{i,1}$. The networks are trained for T=15 (left) and T=30 (right), and the graph displays the PSNR after each intermediate projection. Both graphs share the same y-axis scale.

**Figure 7 entropy-27-00929-f007:**
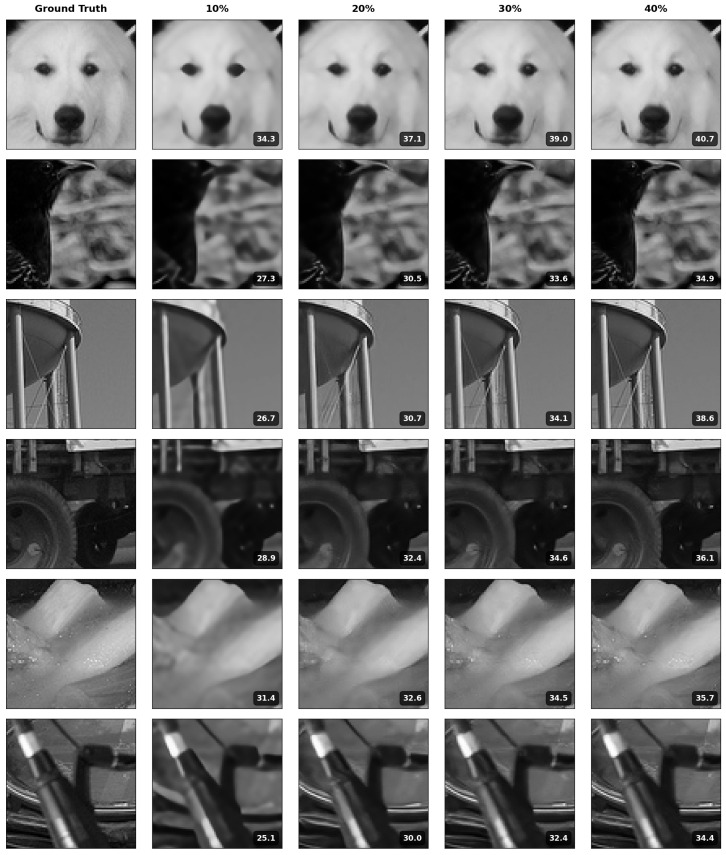
Examples of recovered images (80 × 80) under DCT matrix with 30 projections across different sampling rates. PSNR values shown in dB.

**Table 1 entropy-27-00929-t001:** Average test PSNR (dB) and SSIM ± standard deviation for 30 projection steps across different unrolled algorithms and loss functions ${𝓁}_{ll}$ and ${𝓁}_{i,1}$.

Metric	*m*	Algorithm			
		DeMUN	PGD	Nesterov	AMP
** ${𝓁}_{ll}$ **					
PSNR	0.1n	25.27 ± 5.63	24.17 ± 5.61	24.48 ± 5.55	15.99 ± 2.99
	0.2n	26.37 ± 5.46	27.29 ± 5.96	27.00 ± 5.72	20.02 ± 2.52
	0.3n	28.44 ± 5.21	28.30 ± 5.41	29.24 ± 5.63	23.22 ± 2.82
	0.4n	31.32 ± 5.95	30.33 ± 5.44	29.95 ± 5.46	22.65 ± 2.72
SSIM	0.1n	0.643 ± 0.180	0.590 ± 0.181	0.611 ± 0.183	0.290 ± 0.122
	0.2n	0.697 ± 0.149	0.735 ± 0.144	0.731 ± 0.143	0.436 ± 0.155
	0.3n	0.793 ± 0.111	0.794 ± 0.110	0.817 ± 0.107	0.618 ± 0.109
	0.4n	0.879 ± 0.088	0.850 ± 0.085	0.837 ± 0.093	0.723 ± 0.094
** ${𝓁}_{i,1}$ **					
PSNR	0.1n	26.97 ± 6.42	26.51 ± 6.16	26.19 ± 5.78	26.72 ± 6.26
	0.2n	29.86 ± 6.55	29.06 ± 6.03	28.35 ± 5.66	29.12 ± 6.18
	0.3n	32.05 ± 6.54	30.87 ± 5.85	29.56 ± 5.21	31.24 ± 6.29
	0.4n	34.05 ± 6.71	32.33 ± 5.75	31.14 ± 5.26	32.87 ± 6.21
SSIM	0.1n	0.701 ± 0.180	0.686 ± 0.180	0.678 ± 0.182	0.693 ± 0.184
	0.2n	0.811 ± 0.135	0.796 ± 0.136	0.777 ± 0.140	0.790 ± 0.143
	0.3n	0.873 ± 0.101	0.855 ± 0.104	0.833 ± 0.106	0.855 ± 0.108
	0.4n	0.910 ± 0.075	0.892 ± 0.079	0.874 ± 0.084	0.894 ± 0.083

**Table 2 entropy-27-00929-t002:** Average test PSNR (dB) and SSIM ± standard deviation for 30 projection steps across residual and no residual connections.

Metric	*m*	DeMUN		PGD	
		No Residual	Residual	No Residual	Residual
PSNR	0.1n	26.97 ± 6.42	27.44 ± 6.88	26.51 ± 6.16	26.61 ± 6.77
	0.2n	29.86 ± 6.55	30.74 ± 7.32	29.06 ± 6.03	30.06 ± 6.97
	0.3n	32.05 ± 6.54	32.77 ± 7.09	30.87 ± 5.85	31.88 ± 6.84
	0.4n	34.05 ± 6.71	34.86 ± 7.30	32.33 ± 5.75	33.74 ± 6.81
SSIM	0.1n	0.701 ± 0.180	0.713 ± 0.181	0.686 ± 0.180	0.691 ± 0.178
	0.2n	0.811 ± 0.135	0.824 ± 0.133	0.796 ± 0.136	0.810 ± 0.139
	0.3n	0.873 ± 0.101	0.878 ± 0.102	0.855 ± 0.104	0.857 ± 0.127
	0.4n	0.910 ± 0.075	0.915 ± 0.075	0.892 ± 0.079	0.902 ± 0.080

**Table 3 entropy-27-00929-t003:** Average test PSNR (dB) and SSIM ± standard deviation for 30 projection steps across different loss functions.

Loss Function	Sampling Rate (*m*)			
	0.1*n*	0.2*n*	0.3*n*	0.4*n*
PSNR				
${𝓁}_{i,1}$	27.44 ± 6.88	30.74 ± 7.32	32.77 ± 7.09	34.86 ± 7.30
${𝓁}_{i,0.95}$	27.41 ± 6.78	30.63 ± 7.12	32.92 ± 7.23	34.34 ± 6.93
${𝓁}_{i,0.85}$	27.43 ± 6.95	30.47 ± 7.08	32.79 ± 7.20	34.79 ± 7.22
${𝓁}_{i,0.75}$	27.22 ± 6.61	30.62 ± 7.15	32.58 ± 6.92	34.55 ± 6.94
${𝓁}_{i,0.5}$	27.04 ± 6.58	30.06 ± 6.80	32.23 ± 6.74	33.73 ± 6.70
${𝓁}_{i,0.25}$	26.99 ± 6.62	29.92 ± 6.80	31.97 ± 6.66	33.43 ± 6.56
${𝓁}_{i,0.1}$	26.81 ± 6.67	29.28 ± 6.39	31.79 ± 6.72	33.04 ± 6.35
${𝓁}_{i,0.01}$	26.65 ± 6.31	29.63 ± 6.58	31.65 ± 6.64	33.50 ± 6.61
${𝓁}_{s,5}$	27.31 ± 6.61	30.52 ± 6.98	32.72 ± 6.99	34.71 ± 7.07
${𝓁}_{ll}$	26.75 ± 6.45	23.10 ± 4.14	29.24 ± 5.45	33.69 ± 6.69
SSIM				
${𝓁}_{i,1}$	0.713 ± 0.181	0.824 ± 0.133	0.878 ± 0.102	0.915 ± 0.075
${𝓁}_{i,0.95}$	0.715 ± 0.182	0.825 ± 0.133	0.881 ± 0.100	0.902 ± 0.102
${𝓁}_{i,0.85}$	0.713 ± 0.180	0.821 ± 0.132	0.879 ± 0.099	0.916 ± 0.075
${𝓁}_{i,0.75}$	0.707 ± 0.180	0.824 ± 0.134	0.877 ± 0.099	0.913 ± 0.076
${𝓁}_{i,0.5}$	0.703 ± 0.178	0.811 ± 0.138	0.873 ± 0.100	0.901 ± 0.086
${𝓁}_{i,0.25}$	0.697 ± 0.180	0.806 ± 0.135	0.870 ± 0.101	0.902 ± 0.080
${𝓁}_{i,0.1}$	0.692 ± 0.183	0.800 ± 0.134	0.866 ± 0.105	0.901 ± 0.082
${𝓁}_{i,0.01}$	0.690 ± 0.183	0.804 ± 0.136	0.861 ± 0.105	0.903 ± 0.079
${𝓁}_{s,5}$	0.712 ± 0.184	0.823 ± 0.134	0.878 ± 0.099	0.914 ± 0.079
${𝓁}_{ll}$	0.694 ± 0.180	0.546 ± 0.123	0.809 ± 0.105	0.904 ± 0.076

**Table 4 entropy-27-00929-t004:** Average test PSNR (dB) and SSIM ± standard deviation for 5 projection steps across different network depths.

Metric	*m*	Network Depth (*L*)			
		L=3	L=5	L=10	L=15
PSNR	0.1n	26.33 ± 6.62	26.30 ± 6.48	26.28 ± 6.64	25.99 ± 6.31
	0.2n	29.03 ± 6.69	29.17 ± 6.74	29.27 ± 6.78	28.95 ± 6.44
	0.3n	30.72 ± 6.50	30.95 ± 6.58	31.15 ± 6.74	30.88 ± 6.43
	0.4n	32.22 ± 6.49	32.52 ± 6.65	32.67 ± 6.65	32.49 ± 6.30
SSIM	0.1n	0.673 ± 0.179	0.666 ± 0.182	0.674 ± 0.176	0.665 ± 0.176
	0.2n	0.782 ± 0.135	0.788 ± 0.135	0.793 ± 0.134	0.789 ± 0.133
	0.3n	0.842 ± 0.106	0.847 ± 0.105	0.852 ± 0.103	0.851 ± 0.098
	0.4n	0.881 ± 0.083	0.885 ± 0.080	0.889 ± 0.078	0.888 ± 0.079

**Table 5 entropy-27-00929-t005:** Average test PSNR (dB) and SSIM ± standard deviation for 15 projection steps across different network depths.

Metric	*m*	Network Depth (*L*)			
		L=3	L=5	L=10	L=15
PSNR	0.1n	27.15 ± 6.87	27.22 ± 6.77	27.28 ± 6.85	27.38 ± 6.94
	0.2n	30.06 ± 6.94	30.33 ± 7.11	30.34 ± 6.94	30.19 ± 6.63
	0.3n	32.39 ± 7.09	32.70 ± 7.28	32.67 ± 7.20	32.63 ± 7.00
	0.4n	34.49 ± 7.22	34.43 ± 7.11	34.44 ± 7.16	34.29 ± 6.80
SSIM	0.1n	0.701 ± 0.182	0.708 ± 0.181	0.711 ± 0.179	0.710 ± 0.180
	0.2n	0.807 ± 0.139	0.811 ± 0.142	0.820 ± 0.131	0.818 ± 0.129
	0.3n	0.872 ± 0.102	0.874 ± 0.104	0.878 ± 0.099	0.880 ± 0.098
	0.4n	0.912 ± 0.076	0.908 ± 0.081	0.911 ± 0.079	0.914 ± 0.073

**Table 6 entropy-27-00929-t006:** Average test PSNR (dB) and SSIM ± standard deviation for 30 projection steps across different network depths.

Metric	*m*	Network Depth (*L*)			
		L=3	L=5	L=10	L=15
PSNR	0.1n	27.43 ± 7.00	27.44 ± 6.88	27.51 ± 6.87	27.39 ± 6.99
	0.2n	30.32 ± 6.97	30.74 ± 7.32	30.70 ± 7.06	30.61 ± 7.05
	0.3n	32.67 ± 7.17	32.77 ± 7.09	32.87 ± 7.10	32.74 ± 6.97
	0.4n	34.44 ± 6.99	34.86 ± 7.30	34.70 ± 7.14	34.95 ± 7.34
SSIM	0.1n	0.710 ± 0.180	0.713 ± 0.181	0.714 ± 0.182	0.712 ± 0.182
	0.2n	0.812 ± 0.138	0.824 ± 0.133	0.829 ± 0.132	0.827 ± 0.132
	0.3n	0.874 ± 0.105	0.878 ± 0.102	0.882 ± 0.099	0.882 ± 0.099
	0.4n	0.912 ± 0.075	0.915 ± 0.075	0.913 ± 0.078	0.918 ± 0.075

**Table 7 entropy-27-00929-t007:** Average Test PSNR (dB) and SSIM ± standard deviation across different projection steps and sampling matrices.

*m*	Matrix	Projection Steps		
		5 Steps	15 Steps	30 Steps
PSNR				
0.1n	Gaussian	26.30 ± 6.48	27.22 ± 6.77	27.44 ± 6.88
	DCT	28.42 ± 7.23	28.47 ± 7.19	28.52 ± 7.23
0.2n	Gaussian	29.17 ± 6.74	30.33 ± 7.11	30.74 ± 7.32
	DCT	30.00 ± 7.23	30.37 ± 7.37	30.48 ± 7.48
0.3n	Gaussian	30.95 ± 6.58	32.70 ± 7.28	32.77 ± 7.09
	DCT	31.50 ± 7.11	32.14 ± 7.49	32.20 ± 7.47
0.4n	Gaussian	32.52 ± 6.65	34.43 ± 7.11	34.86 ± 7.30
	DCT	33.22 ± 7.02	33.90 ± 7.44	34.04 ± 7.44

**Table 8 entropy-27-00929-t008:** Average test PSNR (dB) and SSIM ± standard deviation for 30 projection steps across different sampling matrices and noise levels.

Matrix	*m*	Noise Level (σ)			
		0.01	0.025	0.05	0.10
PSNR					
Gaussian	0.1n	27.08 ± 6.46	26.17 ± 5.74	24.89 ± 5.05	23.19 ± 4.54
	0.2n	29.63 ± 6.18	28.50 ± 5.57	26.71 ± 4.89	24.74 ± 4.56
	0.3n	31.58 ± 6.09	29.71 ± 5.15	27.78 ± 4.69	25.60 ± 4.45
	0.4n	32.82 ± 5.76	30.70 ± 4.88	28.63 ± 4.53	26.27 ± 4.43
DCT	0.1n	28.34 ± 6.91	27.97 ± 6.50	27.36 ± 5.95	26.35 ± 5.35
	0.2n	30.05 ± 6.87	29.16 ± 6.14	28.21 ± 5.72	26.73 ± 5.14
	0.3n	31.44 ± 6.58	30.31 ± 5.89	28.91 ± 5.42	27.14 ± 5.00
	0.4n	32.91 ± 6.27	31.32 ± 5.49	29.58 ± 5.02	27.56 ± 4.84
SSIM					
Gaussian	0.1n	0.703 ± 0.180	0.679 ± 0.172	0.636 ± 0.171	0.567 ± 0.180
	0.2n	0.803 ± 0.136	0.782 ± 0.131	0.722 ± 0.139	0.642 ± 0.159
	0.3n	0.865 ± 0.099	0.825 ± 0.103	0.768 ± 0.121	0.684 ± 0.144
	0.4n	0.893 ± 0.077	0.858 ± 0.083	0.802 ± 0.102	0.719 ± 0.130
DCT	0.1n	0.781 ± 0.156	0.771 ± 0.155	0.752 ± 0.154	0.710 ± 0.158
	0.2n	0.825 ± 0.130	0.809 ± 0.129	0.780 ± 0.134	0.727 ± 0.145
	0.3n	0.863 ± 0.104	0.842 ± 0.105	0.806 ± 0.114	0.745 ± 0.132
	0.4n	0.894 ± 0.081	0.871 ± 0.083	0.830 ± 0.094	0.764 ± 0.122

**Table 9 entropy-27-00929-t009:** Test input SNR (dB) under different sampling rates and noise levels.

*m*	Matrix	σ=0.01	σ=0.025	σ=0.05	σ=0.10
0.1n	Gaussian	32.19	24.23	18.21	12.19
	DCT	42.61	34.65	28.63	22.61
0.2n	Gaussian	32.55	24.59	18.57	12.55
	DCT	39.61	31.65	25.63	19.61
0.3n	Gaussian	32.57	24.62	18.59	12.57
	DCT	37.87	29.91	23.89	17.87
0.4n	Gaussian	32.49	24.53	18.51	12.49
	DCT	36.63	28.67	22.65	16.63

**Table 10 entropy-27-00929-t010:** Average test PSNR (dB) and SSIM ± standard deviation for 30 projection steps across different networks depths with DCT matrices.

*m*	Network Depth (*L*)			
	L=3	L=5	L=10	L=15
PSNR				
0.1n	28.51 ± 7.29	28.52 ± 7.23	28.52 ± 7.22	28.51 ± 7.17
0.2n	30.34 ± 7.39	30.48 ± 7.48	30.52 ± 7.46	30.36 ± 7.35
0.3n	32.18 ± 7.61	32.20 ± 7.47	32.18 ± 7.40	31.98 ± 7.29
0.4n	33.88 ± 7.44	34.04 ± 7.44	34.04 ± 7.44	33.91 ± 7.21
SSIM				
0.1n	0.783 ± 0.157	0.785 ± 0.157	0.785 ± 0.157	0.785 ± 0.157
0.2n	0.829 ± 0.132	0.831 ± 0.131	0.832 ± 0.131	0.830 ± 0.131
0.3n	0.869 ± 0.107	0.869 ± 0.105	0.871 ± 0.105	0.868 ± 0.104
0.4n	0.903 ± 0.083	0.903 ± 0.082	0.906 ± 0.081	0.906 ± 0.081

**Table 11 entropy-27-00929-t011:** Average test PSNR (dB) and SSIM ± standard deviation for 30 projection steps across different networks depths and noise levels with Gaussian matrices.

Metric	σ	*L*	Sampling Rate (*m*)			
			0.1n	0.2n	0.3n	0.4n
PSNR	0.01	3	26.95 ± 6.50	29.54 ± 6.18	31.42 ± 6.04	32.81 ± 5.75
		5	27.08 ± 6.46	29.63 ± 6.18	31.58 ± 6.09	32.82 ± 5.76
		10	27.03 ± 6.38	30.06 ± 6.50	31.82 ± 6.27	33.06 ± 5.89
	0.025	3	26.03 ± 5.68	28.30 ± 5.47	29.61 ± 5.16	30.70 ± 4.94
		5	26.17 ± 5.74	28.50 ± 5.57	29.71 ± 5.15	30.70 ± 4.88
		10	26.19 ± 5.65	28.59 ± 5.60	29.98 ± 5.45	30.93 ± 5.04
	0.05	3	24.81 ± 5.04	26.59 ± 4.89	27.83 ± 4.86	28.49 ± 4.49
		5	24.89 ± 5.05	26.71 ± 4.89	27.78 ± 4.69	28.63 ± 4.53
		10	25.03 ± 5.17	26.93 ± 5.13	28.11 ± 5.03	28.71 ± 4.67
SSIM	0.01	3	0.698 ± 0.178	0.805 ± 0.132	0.860 ± 0.099	0.895 ± 0.075
		5	0.703 ± 0.180	0.803 ± 0.136	0.865 ± 0.099	0.893 ± 0.077
		10	0.700 ± 0.180	0.818 ± 0.130	0.869 ± 0.098	0.899 ± 0.075
	0.025	3	0.677 ± 0.174	0.773 ± 0.135	0.823 ± 0.104	0.856 ± 0.086
		5	0.679 ± 0.172	0.782 ± 0.131	0.825 ± 0.103	0.858 ± 0.083
		10	0.686 ± 0.171	0.784 ± 0.131	0.833 ± 0.104	0.864 ± 0.083
	0.05	3	0.633 ± 0.171	0.718 ± 0.138	0.771 ± 0.119	0.798 ± 0.102
		5	0.636 ± 0.171	0.722 ± 0.139	0.768 ± 0.121	0.802 ± 0.102
		10	0.641 ± 0.174	0.728 ± 0.144	0.781 ± 0.119	0.804 ± 0.102

**Table 12 entropy-27-00929-t012:** Average test PSNR (dB) and SSIM ± standard deviation for 30 projection steps across different network depths and noise levels with DCT matrices.

Metric	σ	*L*	Sampling Rate (*m*)			
			0.1n	0.2n	0.3n	0.4n
PSNR	0.01	3	28.32 ± 6.94	29.94 ± 6.84	31.42 ± 6.65	32.79 ± 6.33
		5	28.34 ± 6.91	30.05 ± 6.87	31.44 ± 6.58	32.91 ± 6.27
		10	28.38 ± 6.98	30.10 ± 6.94	31.53 ± 6.69	33.07 ± 6.51
	0.025	3	27.97 ± 6.52	29.08 ± 6.13	30.25 ± 5.94	31.26 ± 5.55
		5	27.97 ± 6.50	29.16 ± 6.14	30.31 ± 5.89	31.32 ± 5.49
		10	27.98 ± 6.53	29.32 ± 6.36	30.37 ± 5.98	31.45 ± 5.59
	0.05	3	27.34 ± 5.96	28.13 ± 5.68	28.88 ± 5.46	29.57 ± 5.13
		5	27.36 ± 5.95	28.21 ± 5.72	28.91 ± 5.42	29.58 ± 5.02
		10	27.37 ± 5.97	28.25 ± 5.77	28.91 ± 5.40	29.71 ± 5.17
SSIM	0.01	3	0.780 ± 0.157	0.823 ± 0.131	0.861 ± 0.106	0.893 ± 0.082
		5	0.781 ± 0.156	0.825 ± 0.130	0.863 ± 0.104	0.894 ± 0.081
		10	0.782 ± 0.156	0.827 ± 0.130	0.864 ± 0.103	0.897 ± 0.080
	0.025	3	0.770 ± 0.155	0.806 ± 0.130	0.840 ± 0.107	0.869 ± 0.086
		5	0.771 ± 0.155	0.809 ± 0.129	0.842 ± 0.105	0.871 ± 0.083
		10	0.772 ± 0.155	0.811 ± 0.129	0.844 ± 0.105	0.873 ± 0.084
	0.05	3	0.750 ± 0.154	0.779 ± 0.133	0.804 ± 0.116	0.827 ± 0.100
		5	0.752 ± 0.154	0.780 ± 0.134	0.806 ± 0.114	0.830 ± 0.094
		10	0.751 ± 0.154	0.783 ± 0.133	0.807 ± 0.112	0.832 ± 0.098

**Table 13 entropy-27-00929-t013:** Average test PSNR (dB) and SSIM ± standard deviation for 30 projection steps across different image resolutions and sampling matrices.

Matrix	*m*	Image Size			
		32×32	50×50	64×64	80×80
PSNR					
Gaussian	0.1n	27.20 ± 7.38	27.44 ± 6.88	28.18 ± 7.21	28.31 ± 6.96
	0.2n	29.78 ± 7.08	30.74 ± 7.32	30.91 ± 6.92	31.54 ± 7.23
	0.3n	32.26 ± 7.44	32.77 ± 7.09	33.22 ± 7.21	33.86 ± 7.32
	0.4n	33.70 ± 7.21	34.86 ± 7.30	34.99 ± 6.91	35.55 ± 7.06
DCT	0.1n	28.70 ± 7.70	28.52 ± 7.23	28.64 ± 7.12	28.83 ± 7.09
	0.2n	30.49 ± 7.84	30.48 ± 7.48	30.85 ± 7.29	31.14 ± 7.42
	0.3n	31.61 ± 7.58	32.20 ± 7.47	32.57 ± 7.40	33.35 ± 7.75
	0.4n	33.46 ± 7.67	34.04 ± 7.44	34.36 ± 7.48	35.01 ± 7.69
SSIM					
Gaussian	0.1n	0.699 ± 0.193	0.713 ± 0.181	0.728 ± 0.181	0.734 ± 0.176
	0.2n	0.806 ± 0.140	0.824 ± 0.133	0.829 ± 0.129	0.831 ± 0.134
	0.3n	0.869 ± 0.108	0.878 ± 0.102	0.878 ± 0.107	0.891 ± 0.095
	0.4n	0.902 ± 0.088	0.915 ± 0.075	0.918 ± 0.071	0.919 ± 0.072
DCT	0.1n	0.777 ± 0.166	0.785 ± 0.157	0.790 ± 0.154	0.796 ± 0.151
	0.2n	0.826 ± 0.138	0.831 ± 0.131	0.830 ± 0.133	0.841 ± 0.124
	0.3n	0.858 ± 0.115	0.869 ± 0.105	0.872 ± 0.104	0.885 ± 0.099
	0.4n	0.897 ± 0.088	0.903 ± 0.082	0.908 ± 0.079	0.914 ± 0.077

## Data Availability

The accompanying code can be found at https://github.com/YuxiChen25/Memory-Net-Inverse (accessed on 9 August 2025).

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
