# Peer review of "Comprehensive Examination of Unrolled Networks for Solving Linear Inverse Problems†"

_entropy, 2025, doi:10.3390/e27090929_

Round 1

Reviewer 1 Report

Comments and Suggestions for Authors

This manuscript  presents a comprehensive examination of unrolled networks for solving linear inverse problems. The work is meaningful and helpful although it is not very innovative.

  1. In this work, only Gaussian and DCT are used as the measurment matrixex, more measurment matrixes should be tested.
  2. The y-axis labels of the right graphs in Figure 4 to Figure 7 should be presented.
  3. Explain why the PSNRs are lower for 30 projections than those for 15 projections.
  4. The performance of the networks are only evaluted by the PSNR metric, more metrics should be calculated to show the superiority of the DeMUN. Some recovered images by the proposed methos should be shown in addtion to the PSNRs results.
  5. The proposed method should be applied to some real scenarios , for expample, to the sparse reconstruction of  CT or MRI images.

Reviewer 2 Report

Comments and Suggestions for Authors

In the proposed manuscript we find a description of the results of a comprehensive empirical study on the design choices for unrolled networks in solving linear inverse problems. The topic is  relevant for various imaging applications. The main question addressed by the research is how to improve the effectiveness of unrolled networks for solving imaging inverse problems.

The main goal of the authors of the manuscript is to solve some of the significant challenges in the field, stemming from the multitude of design choices and the robustness to noise, measurement matrix, and image resolution.

To this end, the authors introduced a deep memory unrolled network, that leverages the history of all gradients and generalizes a wide range of existing unrolled networks. Through extensive simulations, they demonstrated that training deep memory unrolled network with an unweighted intermediate loss function and incorporating residual connections represents the best existing practice for optimizing these networks. Moreover, the authors presented experiments that exhibit the robustness of their design choices to a wide range of conditions, including different measurement matrices, additive noise levels, and image resolutions. The results obtained offer practical guidelines and rules of thumb for selecting the loss function for training, structuring the unrolled network, determining the required number of projections, and deciding on the appropriate number of layers.

The presentation of the main results is clear and comprehensive. The results are valuable and worthy of being published considering their possible development and applications in designing effective unrolled networks for linear inverse problems across various settings.

Minor revisions are suggested to improve the quality of the exposition:

p. 1, line 26: I suggest a period be added at the end of the equation.

p. 3, line 113: I suggest a brief description of the content of the remaining sections to be added at the end of the introductory section

p. 6, line 239: I suggest changing the period with comma after the equations (6).

Reviewer 3 Report

Comments and Suggestions for Authors

This paper addresses the complexity of designing unrolled networks, which are widely used in computer vision and imaging tasks. They focus on the numerous design choices (e.g., optimization algorithm, loss function, network depth) that affect performance and whose fine-tuning is time-consuming, especially when adapting unrolled algorithms to new applications.
The authors suggest a unified parameterized framework (corresponding to some common algorithms under suitable parameter settings) to reduce the design burden and present a detailed ablation study analyzing the impact of each choice, offering practical guidelines. 
This study is of high interest for the scientific community, in my opinion.

The main weakness lies in the experimental evaluation and in the presentation of the results. While the manuscript is well-structured and the experimental pipeline is clearly described, Sections 4 and 5 are overly long and difficult to digest, relying on a lengthy list of 32 tables. The experimental design and result presentation could be streamlined to enhance readability and better guide the reader through the findings.
Moreover, the reported metrics are limited to average PSNR values, which provide only a partial view of the performance. I strongly recommend including standard deviations alongside the averages, as well as complementary quality metrics (e.g., SSIM) to provide a more comprehensive and robust evaluation of the proposed unrolled methods.
Finally, since the experiments are based on the ILSVRC2012 dataset, the paper would benefit from including a selection of visual results. Side-by-side image comparisons would allow readers to qualitatively assess the visual improvements offered by the proposed approach over competing methods.

Minor remarks:

  • The use of the word "LOSS" is misleading. While it is correct for Equations 3-5, in the diagram of Figure 1 the authors should talk about Cost/Objective function (referring to Equation 1).
  • In the Introduction (lines 43-59), the authors should also mention that the considered methods do not solve the associated inverse problem. there is a wide body of literature focusing on this issue (see Sidky, Emil Y., et al. "Do CNNs solve the CT inverse problem?." IEEE Transactions on Biomedical Engineering (2020) for instance)
  • In line 131 the role of T is mentioned, but it is better explained in lines 178-182. Additionally, it is not clear if the T projections are executed by the same trained network or with some trained image-to-image operators.
  • Section 2 must be enriched with references to papers using the cited strategies.
  • Consider moving/repeating lines 644-647 from the Appendix to the main text (section 3 maybe), as they are informative for readers. 
  • At the beginning of Section 4.3 it is not clear if "5 projection steps" means the authors have unrolled T=5 iterations. 
  • Throughout the paper, the authors frequently refer to “the above results” when discussing findings. For clarity and to ensure consistency—especially in case of layout changes during publication—it would be preferable to reference specific table and figure numbers directly. 

Round 2

Reviewer 1 Report

Comments and Suggestions for Authors The paper has been revised as suggested.

Author Response

Thank you so much for your check again!

Reviewer 3 Report

Comments and Suggestions for Authors

The authors have precisely answered my minor observations, but they did not make the major revision I asked for in Sections 4 and 5.  

Author Response

We sincerely thank the reviewer for the suggestions. We greatly appreciate the reviewer's constructive feedback and have undertaken extensive revisions accordingly. We have (1) streamlined our presentation to include only the essential 13 tables and 6 figures that directly support our core narrative, removing all extraneous material, and (2) significantly reduced the paper length to enhance clarity and readability. Each remaining table preserves complete information (PSNR, SSIM, ± SD) to ensure thoroughness while maintaining conciseness.

Round 3

Reviewer 3 Report

Comments and Suggestions for Authors

the paper has been argely improved!

Author Response

We sincerely thank the reviewer for their constructive feedback in helping improve our manuscript!